

# Grazing effects on woody and herbaceous plant biodiversity on a limestone mountain in northern Tunisia

David Anthony Kirk[1,4], Katherine Hébert[2] and Frank Barrie Goldsmith[3]

[1] Department of Geography, University College London, University of London, London, United Kingdom
[2] Département de biologie, Faculté des Sciences, Université de Sherbrooke, Sherbrooke, Québec, Canada
[3] Department of Biology, University College London, University of London, London, United Kingdom
[4] Current affiliation: Aquila Conservation & Environment Consulting, Ottawa, Ontario, Canada

## ABSTRACT

Mediterranean maquis vegetation is highly biodiverse, but widespread grazing poses a challenge for management and conservation. We sampled woody and herbaceous plants separately on a limestone mountain with strong mesic-xeric gradients in Tunisia's Parc National de L'Ichkeul, assessed grazing pressure (on a scale of 1–3), and asked whether grazing had a significant effect on plant compositional abundance before and after controlling for environmental covariates. Sites on the more mesic lakeside face of the mountain were most compositionally unique, and forbs contributed most to the herbaceous beta-diversity on the mountain. We used variance partitioning to separate the collective and individual effects of the abiotic environment, grazing, human activity, and space on herbaceous and woody beta-diversity. However, the individual effect of grazing on overall plant community composition was confounded with space, due to the spatially autocorrelated grazing pressure on the mountain. Importantly, we found that herbaceous and woody communities responded differently to increasing levels of grazing intensity: herbaceous beta-diversity was highest between sites with no grazing pressure, while woody beta-diversity peaked under light grazing. Herbaceous community composition was sensitive to any intensity of grazing pressure, and biotic homogenization occured under moderate-to-high grazing pressure. On the other hand, woody community composition remained relatively similar under no to light grazing pressure, but differed under moderate-to-heavy grazing. Using a one-way permutational analysis of variance analysis, we showed that grazing had a significant effect when controlling for abiotic and spatial covariates. Our findings offer insight into the effects of grazing on maquis vegetation at Jebel Ichkeul, acting as a microcosm of similar conservation and management issues elsewhere in the Mediterranean. We suggest that a combination of monitoring and carefully controlled grazing may enhance plant diversity and maintain the region's biodiverse maquis vegetation, potentially maintaining a key climate refugium for vulnerable endemic species. Importantly, our study provides a useful baseline of the plant assemblages at Jebel Ichkeul with which to compare future vegetation changes.

Corresponding author
David Anthony Kirk,
david@aquilaecology.com

## INTRODUCTION

Humans and their livestock have influenced Mediterranean shrublands for millennia (*Naveh, 1990*). This long history of widespread human activity has played an integral and pervasive role in shaping the region's disturbed, yet remarkably biodiverse landscape (*Papanastasis, 1998*; *Rundel, 1998*; *Mazzoleni et al., 2004*; *Vogiatzakis, Mannion & Griffiths, 2006*; *Blondel et al., 2010*). Evaluating the impacts of humans' coevolution with the Mediterranean shrublands is thus an ongoing challenge for the management and conservation of this region's biodiversity (*Falcucci, Maiorano & Boitani, 2007*; *Rundel et al., 2016*).

Historically, low intensity grazing by domestic livestock and fires set by shepherds maintained the floristically rich and heterogeneous landscapes of the Mediterranean maquis, which were previously shaped by more abundant native herbivores (*Le Houerou, 1981*; *Papanastasis, 1998*; *Papanastasis, Kyriakakis & Kazakis, 2002*). The resulting shrubland communities are of high ecological value, due to their rich diversity of annuals, and geophytes from the Orchidaceae, Iridaceae, and Liliaceae (*Pons, 1981*; *Quézel, 1981*). Under certain conditions, grazing can also effectively restore plant diversity after shrub encroachment in dry grasslands (*Elias & Tischew, 2016*; *Elias, Hölzel & Tischew, 2018*; see also *Török et al., 2016*; *Török et al., 2018*) and in heathlands (*Rupprecht, Gilhaus & Hölzel, 2016*). Without such disturbance, many shrublands develop into forests, leading to a loss of plant diversity and a build-up of organic matter, rendering them susceptible to wildfires (*Rackham & Moody, 1996*; *Perevolotsky & Seligman, 1998*; *Henkin, 2011*). Although grazing can maintain endemic species and regulate ecosystems (*Gomez-Campo, 1985*), overgrazing and uncontrolled wood-cutting can also reduce plant diversity, and cause erosion or desertification (*Hill et al., 1998*; *Papanastasis, 1998*; *Papanastasis, Kyriakakis & Kazakis, 2002*). Evaluating the effects of different intensities of grazing disturbance on biodiversity is therefore fundamental to inform management strategies in maquis ecosystems.

The effects of grazing on plant biodiversity are equivocal, being subject to various temporal, spatial, and ecological contingencies (*Olff & Ritchie, 1998*; *Olsvig-Whittaker et al., 2006*). On islands, intensive overgrazing by goats has devastated vegetation and threatened the persistence of many island endemics (*Campbell & Donlan, 2005*). However, many Mediterranean plant species have coevolved with herbivores and developed strategies to resist grazing, including spininess, chemical repulsion, prostrate growth, and an ability to grow on remote rocky cliffs that are inaccessible to livestock (*Papanastasis, 1998*). Grazing has therefore applied a continuous selective pressure on plant species over extensive time scales, likely influencing plant community composition and diversity across the entire region (*Rundel et al., 2016*).

Characterized by extremely high floristic richness, the Mediterranean region holds about 10% of global plant species diversity (25,000 species) on only 2% of the world's terrestrial surface area, and half of these plant species are endemic to the region (*Médail & Quézel, 1997*; *Radford, Catullo & De Montmollin, 2011*). The stability of climate across timescales, including provision of Pleistocene refugia, is thought to have prevented species' extinctions and facilitated speciation, leading to this remarkably high biodiversity in the Mediterranean

Basin (*Nogués-Bravo et al., 2008*; *Nogués-Bravo, López-Moreno & Vicente-Serrano, 2012*). The resulting characteristic shrub formations occur only in five areas: California (United States), Chile, South Africa, Australia and the Mediterranean Basin in southern Europe and North Africa (*Tomaselli, 1977*; *Di Castri, 1981*; *Cowling et al., 1996*).

Given the long history of human activity in the Mediterranean basin, implementing suitable management strategies to maintain and protect these biodiversity hotspots is a major conservation dilemma (*Blondel et al., 2010*). Primary "natural" vegetation remains in only 4.7% of the Mediterranean basin (*Geri, Amici & Rocchini, 2010*), and many of these remaining wildlands are restricted to mountainous or coastal regions with steep slopes that preclude cultivation. Some protected areas were implemented to conserve Mediterranean shrublands, but do not cover enough land to effectively buffer them against the erosion of vegetation diversity (*Wilson et al., 2007*). With forecasted increases in the frequency and magnitude of droughts and desertification due to climate change, areas of high plant endemism in the Mediterranean will likely lose their biodiversity and functional heterogeneity (*De la Riva et al., 2016*; *Gauquelin et al., 2016*). Thus, the challenge of integrating multiple uses of Mediterranean landscapes to effectively protect the region's remarkable diversity is fundamental to the conservation of Mediterranean flora.

Jebel Ichkeul, a limestone mountain within Le Parc National de L'Ichkeul in the north of the Republic of Tunisia, represents a microcosm of the conservation issues faced by Mediterranean vegetation, including grazing. In a recent assessment of the threats to Important Plant Areas (IPAs) in the southern and eastern Mediterranean, *Radford, Catullo & De Montmollin (2011)* identified overgrazing as the main threat to IPAs in Tunisia (see also *Underwood et al., 2009*). Assessing the influence of grazing pressure and human activity on plant community composition at Jebel Ichkeul can therefore provide insight into how these threats might be affecting other Mediterranean shrublands, and to inform their protection and management.

In this paper, we investigate the influence of grazing pressure on woody and herbaceous plant communities on Jebel Ichkeul, a limestone mountain with strong mesic-xeric gradients. More specifically, we ask: (1) how grazing, the abiotic environment, and space collectively and separately shape herbaceous and woody plant community composition on Jebel Ichkeul, with a special focus on the influence of grazing; (2) how the intensity of human activity (using *O. europaea* size and density as a proxy) influences plant community composition on Jebel Ichkeul, in concert with and independently of these ecological drivers; and (3) how different grazing intensity levels influence plant beta-diversity on Jebel Ichkeul, and which plant taxonomic and functional groups are associated with each level.

We predicted that grazing, as well as the abiotic environment and spatial relationships between sites, would collectively and individually explain variation in plant community composition across sites (*Osem, Perevolotsky & Kigel, 2007*). Second, we expected that human activities like woodcutting would apply a selective pressure on community composition, explaining additional plant beta-diversity across sites (*Agra & Ne'eman, 2009*). Third, we predicted that grazing intensity should influence beta-diversity, where beta-diversity should be highest under light browsing, and should be lowest under both grazing extremes (i.e., unbrowsed and moderate-to-heavily intensity; *Lázaro et al., 2016*).

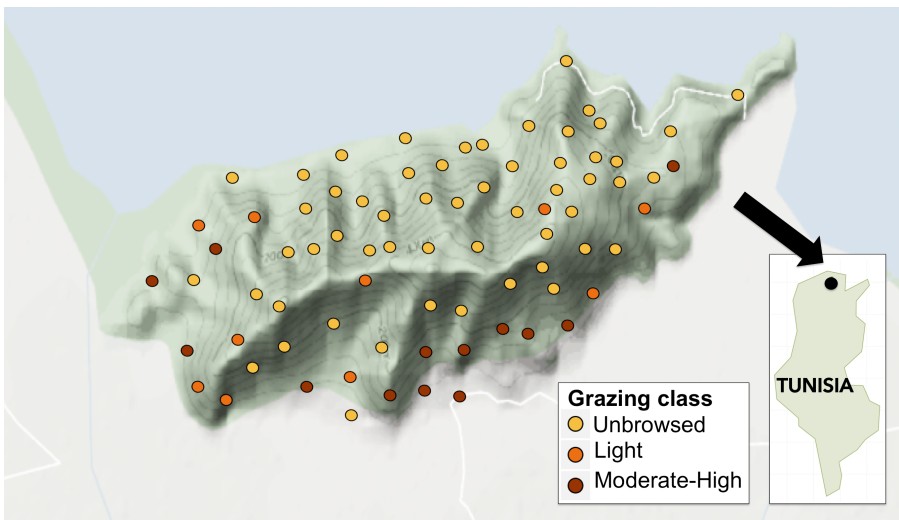

**Figure 1   Map of study sites on Jebel Ichkeul within Le Parc National de L'Ichkeul, and location within Tunisia (inset).** Points are coloured according to grazing class, where yellow, unbrowsed; orange, light browsing; and red, moderate-to-high browsing.

## METHODS

### Study area

Set in the Mateur plain, Jebel Ichkeul is 25 km southwest of Bizerte in northeastern Tunisia and 15 km south of the Mediterranean Sea (37°10′N, 09°40′E; Fig. 1). It is surrounded along its northern flanks by Garaet el Ichkeul, an internationally important wetland; both the Jebel and lake are included within the National Park and World Heritage Site (IUCN, 2003). South of the southern perimeter of the Jebel, there is intensive arable farming, pasture and orchards (*UNEP-WCMC & IUCN, 2017*). Formerly an island within Lac Ichkeul, the dolomitic massif covers an area of 1,363 ha (13.6 km$^2$); 690 ha of this was gazetted in the original declaration of the National Park in 1977 (*Hollis, 1977*; *Hollis, 1986*). Jebel Ichkeul was listed as an Important Plant Area (IPA) in 2000 (*Radford, Catullo & De Montmollin, 2011*), and is nationally important (*Peterken & Radford, 1971*). Of the 29 plant families known to occur in the Mediterranean region (*Quézel, 1981*), 24 occur on the Jebel, including the rare Tunisian endemic *Teucrium schoenenbergeri* (*Fay, 1980*).

Jebel Ichkeul was listed in the European Red List of Habitats as one of the most typical examples of *Olea europaea* var *sylvestris* with *Ceratonia siliqua* and *Pistacia lentiscus* along with southern Andalusia, Menorca, Sardinia, Sicily, Calabria and Crete (*European Environment Agency, 2017*). At the time of our study, the Jebel's forests and maquis provided grazing for livestock, especially in autumn and winter (*Hollis, 1977*). At other times of year, the wetlands and marshes of the National Park were a source of forage for over 2,500 + livestock (Anonymous, 1998, unpublished data). Combined with uncontrolled woodcutting for firewood on the southern slopes, grazing pressure caused loss of vegetation cover and reduced species diversity, leading to increased soil erosion and the spread of invasive plants (*Fay, 1980*).
The vegetation of Jebel Ichkeul is extremely heterogeneous due to its varied geology (dolomite, calc-schist and marble), dissected relief, high altitude (maximum 512 m.a.s.l.), the adjacent Lac Ichkeul, and spatially-variable anthropogenic activity (*Daoud-Bouattour, Gammar Ghrabi & Limam Ben Saad, 2007*). The northern slopes facing Lac Ichkeul are generally mesic with relatively continuous vegetation cover. By contrast, the southern slopes of the mountain feature xeric plant communities, often in various states of degradation. These effects were most apparent close to gourbi village settlements and the Hammams (hot springs) located at the foot of the mountain. Illegal quarries, since closed, also mined limestone on the southern slopes of the Jebel, and contributed to the xeric conditions at these sites, and potentially dust pollution (*Kirk, 1983*). A road running along the southern flanks of the mountain links the villages and quarries and is served by some secondary roads from the Mateur Plain (Fig. 1). On the southern slopes of the mountain, calc-schist parent material results in sites with high soil pH, while the northern slopes are mostly dolomitic. Jebel Ichkeul occurs in the Mediterranean bioclimatic zone with a summer drought (*Daget, 1977*). Mean monthly temperatures range from 11.3 °C in January (winter minimum 0 °C) to a mean of 25.2 °C in July (summer maximum 40 °C). The average annual rainfall is 625 mm, with only 4 per cent of this falling in summer (*Hollis, 1977*).

## Plant community surveys and sampling design

The first author conducted all sampling between 18 June and 7 August 1983. To describe plant species distribution and abundance, we located 78 quadrats on the Jebel, stratified using the data and vegetation maps in *Fay (1980)* for guidance. Except for inaccessible crags and steep cliffs, we sampled all vegetated areas. Although the general areas of survey sites were not chosen randomly, they were widely dispersed and spaced over the Jebel and sampled all community types in *Fay (1980)*. At each site, we threw a quadrat randomly to locate the central stake. We used a nested quadrat design with dimensions of 2 × 2 m, 5 × 5 m, 7.07 × 7.07 m, 10 × 10 m, and 14 × 14 m (*Bunce, 1982*). We estimated cover-abundance for herbaceous and woody species within the 2 × 2 m quadrat and the 10 × 10 m quadrat, respectively, using a modified Braun Blanquet scale (e.g., <1%, 1–5%, 6–10%, 11–25%, 26–50%, 51–75%, >75%). In the remaining quadrats, we recorded presence-absence of species, but here present only data from the 2 × 2 m and 10 × 10 m quadrats. On average, three sites (nested quadrats) were surveyed during 8–10 h per day.

## Identifying plant species and development of functional groups

Plant names and occurrences in the national park were verified by a plant ecologist (A. Daoud Bouattour, University of Tunis, pers. comm., 2012), and final taxonomy is based on a catalogue of plants of Tunisia (*Le Floc'h, Boulos & Vela, 2010*), as well as a flora of the national park compiled since our study (*Daoud-Bouattour, Gammar Ghrabi & Limam Ben Saad, 2007*) and the *Euro-Med Plantbase (2017)*.

Because we surveyed woody and herbaceous species' abundance at different spatial scales, we considered herbaceous and woody plant communities separately in all analyses. We further classified species into taxonomic and functional groups predicted to respond differently to grazing intensity: Poaceae (Graminoids), Legumes, Geophytes, Forbs (annual

and perennial), shrub legumes and shrubs/trees (similar to those in *Fernández-Lugo et al., 2013*). The "Forbs" functional group contains ferns, cactuses, club mosses and some unidentified mosses. We did not differentiate between annual and perennial species as in *Fernández-Lugo et al. (2013)*, because our study took place over a single season.

## Assessing grazing intensity

At each of the quadrat locations within the 10 × 10 m square, we ranked browsing by livestock on the principal woody plant species. We recognized that the effect of grazing could vary between woody and herbaceous species, that certain species might be preferred, and that these preferences may differ by livestock type (*Tóth et al., 2018*). However, we were unable to quantify grazing on herbaceous species, and therefore assumed that browsing on woody plants (probably almost entirely by goats) was indicative of overall grazing intensity on both woody and herbaceous species.

We categorized grazing intensity into three levels: (1) unbrowsed (i.e., dense forests, forests with gaps or high altitude arborescent matorral), (2) lightly browsed (i.e., dense middle or low matorral with trees showing signs of browsing on shrubs), and (3) moderate to heavy browsing (i.e., bushes clipped to a hedge-like shape or severely clipped to short stunted bushes), following *Tomaselli (1977)*. We also recorded the activity of herbivorous mammals qualitatively by signs such as hair on tree trunks, droppings, and feeding signs from wild boars (*Sus scrofa*) such as uprooted tubers and soil disturbance, but these were not quantified. Of the 78 sampled sites, 71% ($n = 55$) were classified as unbrowsed (level 1), 12.8% ($n = 10$) as lightly browsed (level 2), and 16.7% ($n = 13$) as moderately to heavily browsed (level 3). Moderately to heavily grazed sites (level 3) were generally of low altitude with more rock outcropping and smaller-diameter *O. europaea* trees. See Fig. 2 for an assessment of percent occurrence and relative abundance of woody species in each grazing class.

## Measuring abiotic environmental variables

To disentangle the effect of grazing from environmental covariates on plant species compositional turnover (*Arévalo et al., 2011a*; *Arévalo et al., 2011b*), we Arévalo abiotic variables at the centre of each quadrat (Table 1; see below). We recorded altitude with a barometric field altimeter. Because slope aspect plays an important role in the distribution of woody vegetation in the Mediterranean (*Sternberg & Shoshany, 2001*), we also measured slope using an inclinometer (Abney Level—Eugene Dietzgen, Chicago, USA) and recorded compass aspect as a proxy for solar insolation. Parent material characteristics play a strong role in shaping Mediterranean shrub communities (*Molina-Venegas et al., 2016*), so we recorded the percentage of rock outcropping or rock cover at quadrats.

### *Olea europaea* as a proxy for anthropogenic activity

Woody vegetation shades other plant species, and thus modulates the composition and abundance of herbaceous plant species (*Agra & Ne'eman, 2009*; *Segoli et al., 2012*). Woody vegetation, especially *O. europaea*, is also a key indicator and surrogate for the intensity of anthropogenic activities, including grazing and woodcutting. Therefore, we measured *O. europaea* stems (>4 cm) in the 10 × 10 m quadrat at breast height (dbh; c 1.5 m) and

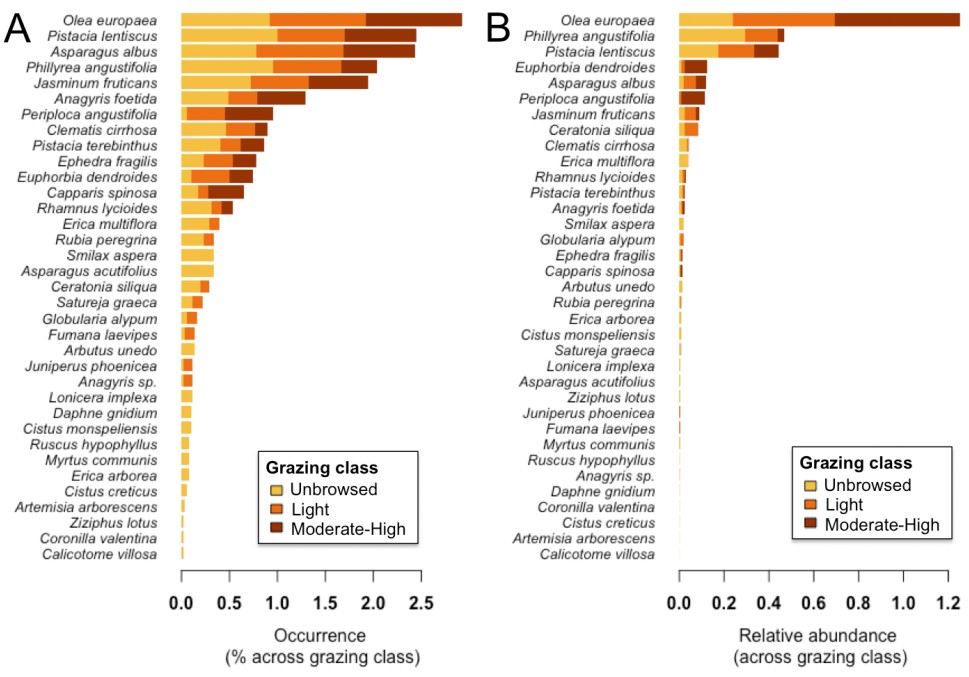

**Figure 2** (A) Percent occurrence and (B) relative abundance of woody plant species on Jebel Ichkeul within each of three grazing classes. Bars are coloured according to grazing class, where yellow, unbrowsed; orange, light browsing; and red, moderate-to-high browsing.

at 30 cm above ground level for 69 sites (data were not available for 9 sites). We present results using stem size at 30 cm height, because most trees were too small to obtain a breast height measurement in shallow soils or heavily grazed sites. We then calculated *O. europaea* densities in 0–5, 6–10, 11–15, 16–20 and >20 cm circumference size classes. Where trees bifurcated, all stems were measured and a calculation used to provide a single diameter comparable with a tree of the same surface area (Supplemental Information 1).

## Spatial autocorrelation between sites

To assess positive spatial autocorrelation in plant community composition across the mountain, we computed the local Moran's I for each site based on Hellinger distances, and tested for significance using 999 random permutations. We also made the assumption that certain unmeasured environmental covariates were spatially autocorrelated. For example, prevailing westerly winds carry moisture from the lake and may create a humidity gradient from northerly mesic vegetation to xeric southern vegetation at Jebel Ichkeul (*Fay, 1980*). Explicitly incorporating spatial relationships between sites can help account for this spatial gradient in humidity and soil moisture, which can have a strong influence on Mediterranean maquis vegetation (*Sardans & Peñuelas, 2013*). To account for the positive spatial autocorrelation between sites in subsequent analyses, we computed positive distance-based Moran's eigenvector maps (dbMEMs; *Dray, Legendre & Peres-Neto, 2006*; *Legendre & Legendre, 2012*; *Legendre & Gauthier, 2014*) based on site coordinates determined from

**Table 1  Biophysical variables measured at Jebel Ichkeul.**

| Variable | Mean ± SE | Range | N |
|---|---|---|---|
| Altitude m.a.s.l. | 143.0 ± 12.7 | 6–404 | 78 |
| Aspect | | 20–360° | 78 |
| Slope | 23.0 ± 1.1 | 4.3-64.5 | 78 |
| Rock (% cover) | 36.9 ± 3.2 | 0–90 | 78 |
| pH | 7.36 ± 0.03 | 6.7–7.7 | 50 |
| Density *Olea* A (0–5 cm) | | | |
| 30 cm | 1.48 ± 0.45 | 0–25 | 69 |
| DBH | 2.03 ± 0.63 | 0–32 | 69 |
| Density *Olea* B (6–10 cm) | | | |
| 30 cm | 2.03 ± 0.42 | 0–15 | 69 |
| DBH | 1.81 ± 0.38 | 0–15 | 69 |
| Density *Olea* C (11–15 cm) | | | |
| 30 cm | 1.41 ± 0.24 | 0–8 | 69 |
| DBH | 1.20 ± 0.21 | 0–7 | 69 |
| Density *Olea* D (16–21 cm) | | | |
| 30 cm | 0.74 ± 0.18 | 0–8 | 69 |
| DBH | 0.39 ± 0.10 | 0–3 | 69 |
| Density *Olea* E (21 cm +) | | | |
| 30 cm | 0.41 ± 0.11 | 0–4 | 69 |
| DBH | 0.20 ± 0.08 | 0–4 | 69 |

Google Earth. Local Moran's I and the positive dbMEMs were calculated using the R package *adespatial* (*Dray et al., 2018*).

## STATISTICAL ANALYSIS

### Beta-diversity of herbaceous and woody communities

We assessed the compositional differences between sites in herbaceous and woody communities as the Hellinger distances between sites for each community type (*Anderson et al., 2011*). To determine which sites harboured more ecologically unique communities, we quantified each site's local contribution to beta-diversity (LCBD) for each community type (*Legendre & De Cáceres, 2013*). To identify the plant groups that most strongly drive beta-diversity between sites, we also assessed each species' contribution to beta-diversity (SCBD), summed for each functional group (Poaceae, legumes, forbs, geophytes; *Legendre & De Cáceres, 2013*). We tested the significance of LCBD values using permutation tests with 999 iterations.

### Disentangling the ecological and anthropogenic drivers of plant beta-diversity

We first determined the combined and separate influences of ecological drivers on plant community structure. We used distance-based redundancy analyses (db-RDA) to quantify the combined influence of the abiotic environment (altitude, slope, aspect, and rock-crop cover), grazing intensity, and spatial autocorrelation (dbMEMs) on the Hellinger

dissimilarities between sites in the herbaceous and the woody plant communities. Model significance tests were based on 999 permutations. We then used variation partitioning to disentangle the individual influences of the abiotic environment, grazing, and spatial autocorrelation on the herbaceous and woody plant beta-diversity (*Borcard, Legendre & Drapeau, 1992*; *Peres-Neto et al., 2006*). For the two plant community types, we partitioned Hellinger dissimilarities into fractions of variation explained by [a] abiotic environmental variables (altitude, slope, aspect, and rock outcrop cover), [b] grazing classes, [c] spatial autocorrelation (dbMEMs). We then tested the significance of each individual fraction using permutation tests with 999 iterations.

To determine the influence of anthropogenic activity on plant community composition, we repeated the db-RDA and variation partitioning analyses including only sites with *O. europaea* sizes and densities. For the two plant community types, we therefore partitioned Hellinger dissimilarities into fractions of variation explained by [a] abiotic environmental variables (altitude, slope, aspect, and rock outcrop cover), [b] grazing classes, [c] human activity (*O. europaea* sizes and densities), and [d] spatial autocorrelation (db-MEMs).

All db-RDA and variation partitioning analyses were conducted using the R package *vegan* (*Oksanen et al., 2019*).

## Influence of grazing intensity on plant beta-diversity

To determine how grazing intensity influences plant beta-diversity, we tested for significant differences in the variance of Hellinger dissimilarities between the three levels of grazing intensity. We computed a test of the multivariate homogeneity of group dispersions between grazing classes and assessed significance of between-group differences using 999 permutations (*Anderson, 2006*). Smaller multivariate site distances to the grazing-specific centroids are interpreted as biotic homogenization, whereas larger distances are interpreted as biotic differentiation. The multivariate homogeneity of group dispersions test was performed using the R package *vegan* *Oksanen et al. (2019)*.

To further tease out the influence of grazing intensity from the effects of abiotic, spatial, and anthropogenic covariates, we performed one-way permutational multivariate analysis of variance (PERMANOVA) models with grazing as a factor and controlling for covariates. We selected abiotic and spatial variables through sequential AIC$_c$ comparison of db-RDA models. We first compared db-RDA models including all abiotic covariates and grazing, and retained abiotic variables from the best model. We repeated this with spatial variables (dbMEMs). A PERMANOVA was then conducted to evaluate grazing intensity's effects on beta-diversity, with the retained abiotic and spatial variables as covariates. Because grazing intensity is spatially structured, we forced the order of entry in the PERMANOVA by entering grazing intensity first in one set of models, and then entering it last in a second set of models. PERMANOVAs were conducted first for all 78 sites, then separately for the subset of 69 sites for which anthropogenic variables (i.e., *O. europaea* densities by size class) were available. We assessed explanatory power using Type I sums of squares in the PERMANOVAs. We performed the db-RDAs (best model subsets) and permutational multivariate analysis of variance using PERMANOVA + (*Anderson, Gorley & Clarke, 2008*).
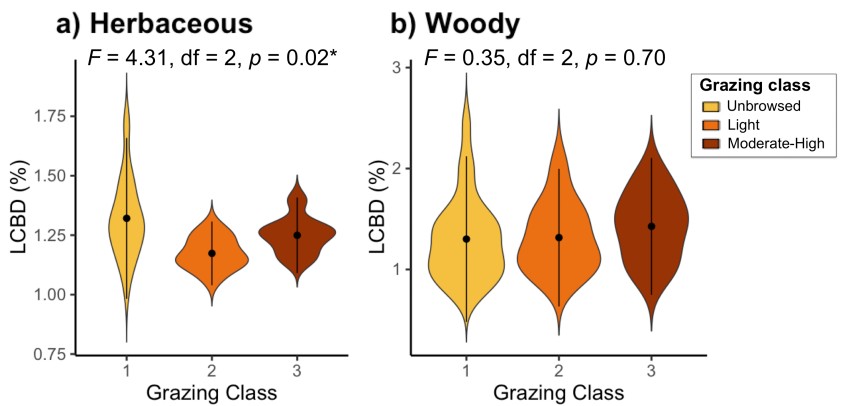

**Figure 3** Local contributions to beta diversity (LCBD) of sites grouped by grazing class for (A) herbaceous and (B) woody plant communities on Jebel Ichkeul.

# RESULTS

## Spatial autocorrelation between sites

Moran's I tests demonstrated positive spatial autocorrelation in plant community composition within grazing classes, particularly in the woody communities (Fig. S1). Herbaceous community composition showed significant positive spatial autocorrelation in 21 of 55 (38.2%) of unbrowsed sites, five of 10 (50%) of lightly-browsed sites, and 11 of 13 (84.6%) of moderately-to-heavily browsed sites. Most strikingly, the sites with the highest positive spatial autocorrelation for herbaceous species were on the southern, heavily grazed slopes of the Jebel (Fig. S1). Woody community composition showed high levels of positive spatial autocorrelation across all grazing classes: 47 out of 55 (85.5%) in unbrowsed sites, nine out of 10 sites (90%) in lightly-browsed sites, and 11 out of 13 (84.6%) in moderately-to-heavily browsed sites. The influence of grazing and of other ecological and anthropogenic processes therefore needed to be disentangled from the pervasive effect of spatial autocorrelation between sites in both community types.

## Beta-diversity of herbaceous and woody communities

Sites with the highest local contribution to beta-diversity (LCBD) (i.e., more compositionally unique sites) were generally located on the lakeside face of the mountain, contributing up to 2.04% (site 44) of total herbaceous beta-diversity, and up to 2.47% (site 51) of the overall woody beta-diversity on the Jebel (Fig. S2). LCBD also differed significantly among grazing classes for herbaceous plant species ($F_2 = 4.31$, $df = 2$, $P = 0.02$), but not for woody species ($F_2 = 0.35$, $df = 2$, $P = 0.7$; Fig. 3).

Forbs contributed most to herbaceous beta-diversity (SCBD), followed by Poaceae, Legumes, and Geophytes (Fig. 4). Woody beta-diversity was largely attributed to woody species other than legumes (Fig. 4). Legumes almost exclusively contributed to beta-diversity in unbrowsed and lightly browsed sites. It is also notable that of the 35 woody species, only 12 occurred in heavily grazed sites (Fig. 2).

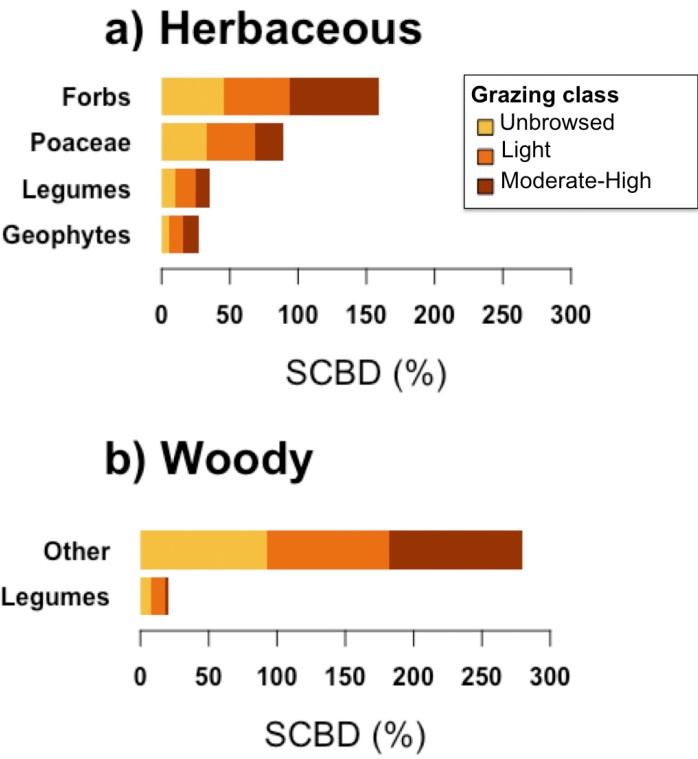

**Figure 4** Species contributions to beta diversity (SCBD) of each functional group, in sites of each grazing class, for (A) herbaceous and (B) woody plant communities.

## Disentangling the ecological and anthropogenic drivers of plant beta-diversity

The importance of ecological drivers on plant beta-diversity differed between herbaceous and woody communities. In herbaceous communities, the db-RDA showed the abiotic environment, grazing intensity, and spatial variables collectively explained relatively little ($R_{adj} = 0.047$, $p = 0.001$) of the variation in beta-diversity between the 78 sites. Variation partitioning indicated that abiotic variables ([a]) significantly explained a small portion of this variation ($R_{adj} = 0.015$, $p = 0.007$), while grazing intensity ([b]) and spatial variables ([c]) showed no significant effects (Fig. 5A). The individual effect of grazing intensity ([b]) could not be partitioned from the effects of abiotic and spatial variables ($R_{adj} = -0.003$, $p = 0.8$). Here, a negative $R_{adj}$ indicated that grazing intensity only explained the variation in beta-diversity when considered with the abiotic and spatial covariates (*Legendre & Legendre, 2012*). This suggests that grazing intensity is highly correlated with the abiotic and spatial variables, making it difficult to isolate its influence on herbaceous community composition.

In the woody community (Fig. 5B), the db-RDA demonstrated that 12.5% ($R_{adj} = 0.125$, $p = 0.001$) of the variation in beta-diversity could be attributed to the combined effect of the abiotic environment, grazing intensity, and spatial variables. As in the herbaceous community, the individual effect of grazing intensity ([b]) could not be partitioned from
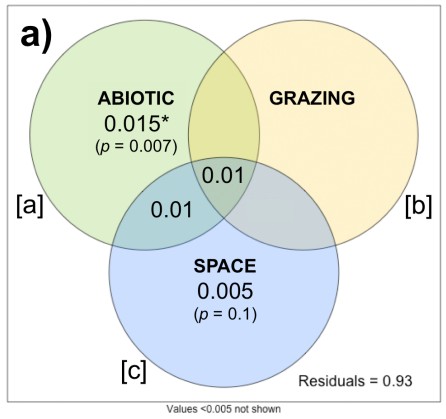
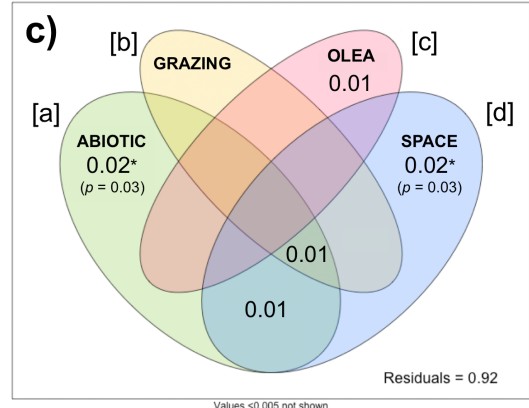

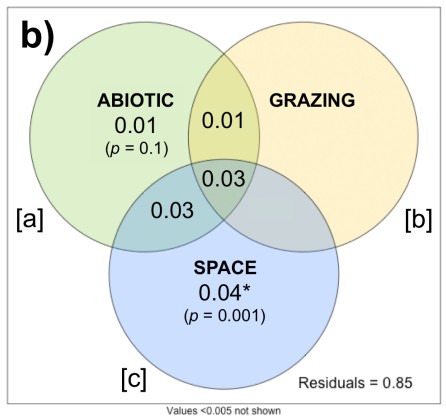
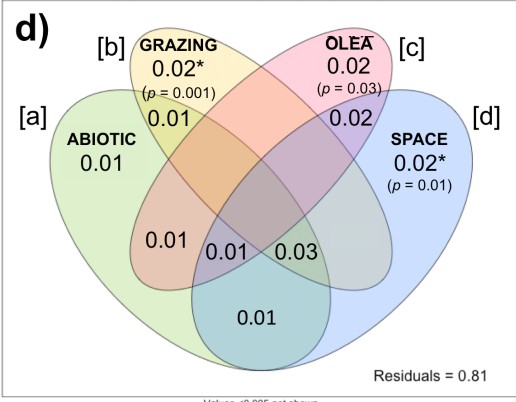

**Figure 5** **Variation partitioning of herbaceous (A, C) and woody (B, D) beta-diversity.** In (A) and (B): beta-diversity explained by abiotic, grazing, and spatial variables across 78 sites. In (C) and (D): beta-diversity explained by abiotic, grazing, *O. europaea* sizes and densities (as proxy for human activity), and spatial variables across 69 sites.

the effects of the abiotic environment and spatial variables ($R_{adj} = -0.005, p = 0.9$). Space alone ([c]) significantly explained the greatest portion of the variation in woody beta-diversity across sites ($R_{adj} = 0.044, p = 0.001$), while the individual effect of the abiotic environment ([a]) did not significantly explain plant beta-diversity after partitioning ($R_{adj} = 0.011, p = 0.11$).

To investigate the influence of anthropogenic activity on plant beta-diversity, we repeated these analyses on 69 sites with the inclusion of *O. europaea* densities, as a proxy for human activities such as woodcutting. Including *O. europaea* density covariates slightly increased the explanatory power of both db-RDA models: the combined effect of the abiotic environment, grazing intensity, *O. europaea* densities, and spatial variables explained 7.6% ($R_{adj} = 0.076, p = 0.001$) of herbaceous beta-diversity, and 16.4% ($R_{adj} = 0.164, p = 0.001$) of woody beta-diversity.

Variation partitioning further discerned the individual effects of the ecological and anthropogenic covariates. In the herbaceous community, space alone ([d]) still explained

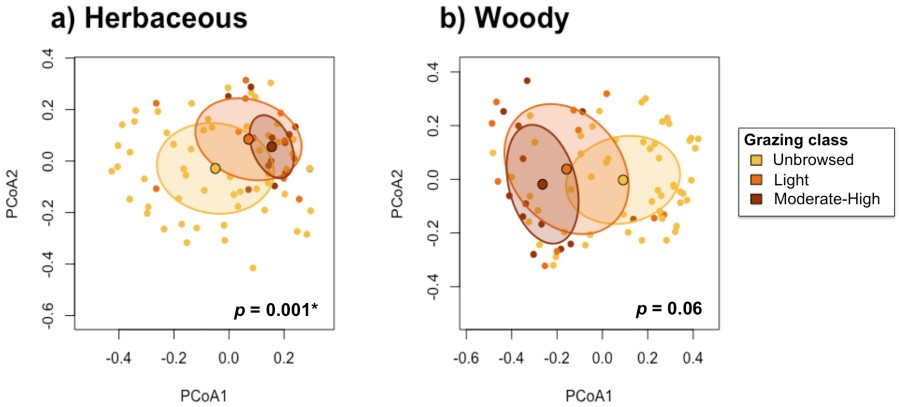

**Figure 6** Test for homogeneity of multivariate dispersions between grazing classes for (A) herbaceous and (B) woody species.

most variation ($R_{adj} = 0.019$, $p = 0.03$), followed closely by the abiotic environment ([a]) ($R_{adj} = 0.021$, $p = 0.03$) (Fig. 5C). The individual effects of grazing intensity ([b]) and *O. europaea* densities ([c]) were both unable to significantly explain variation in herbaceous beta-diversity ([b]: $R_{adj} = 0.001$, $p = 0.4$; [c]: $R_{adj} = 0.012$, $p = 0.09$), although the effect of *O. europaea* densities was significant when considered with all other covariates ($R_{adj} = 0.015$, $p = 0.01$). Importantly, in the woody community, the individual effects of grazing intensity ([b]) and *O. europaea* densities ([c]) were successfully partitioned from the effects of spatial variables and the abiotic environment, respectively explaining 2.2% ($p = 0.001$) and 2.1% ($p = 0.03$) of woody beta-diversity (Fig. 5D). However, spatial variables still explained the largest fraction of woody beta-diversity ($R_{adj} = 0.024$, $p = 0.01$).

## Influence of grazing intensity on plant beta-diversity

The test for the homogeneity of multivariate dispersions indicated that beta-diversity differed significantly among grazing classes in the herbaceous communities ($p < 0.001$), and was marginally significantly different among grazing classes in the woody communities ($p = 0.065$; Fig. 6). Herbaceous communities were more differentiated among unbrowsed sites, and most homogenized in moderately-to-heavily grazed sites (Fig. 6). Although the pattern was less pronounced in the woody community, moderately-to-heavily grazed sites showed more homogenization than sites with less grazing pressure (Fig. 6).

Pairwise tests demonstrated significant differences in herbaceous beta-diversity between unbrowsed and lightly browsed sites ($p < 0.001$), and between unbrowsed and moderately-to-heavily browsed sites ($p < 0.001$). However, herbaceous communities did not appear to differ significantly under light browsing and moderate-to-heavy browsing ($p = 0.98$). Woody beta-diversity differed significantly between unbrowsed and moderately-to-heavily browsed sites ($p = 0.22$). However, beta-diversity did not differ significantly between lightly browsed and unbrowsed sites ($p = 0.55$), or between lightly browsed and moderately-to-heavy browsed sites ($p = 0.16$).
**Table 2  Permutational multivariate analysis of variance (Hellinger distance) testing herbaceous and woody species composition among grazing classes (crossed design, grazing as fixed effect) at 78 sites controlling for abiotic and spatial covariates.** (a) Herbaceous; (b) Woody.

| Source | Df | SS | MS | Pseudo F | P (perm) | # unique permutations |
|---|---|---|---|---|---|---|
| (a) Herbaceous species | | | | | | |
| *Grazing first* | | | | | | |
| Grazing | 2 | 2.6875 | 1.3437 | 1.995 | 0.0003 | 9830 |
| MEM1 | 1 | 1.5919 | 1.5919 | 2.363 | 0.0013 | 9880 |
| Residuals | 74 | 49.845 | 0.6736 | | | |
| Total | 77 | 54.12 | | | | |
| *Grazing last* | | | | | | |
| MEM1 | 1 | 1.5664 | 1.5664 | 2.326 | 0.0006 | 9858 |
| Grazing | 2 | 2.713 | 1.3565 | 2.014 | 0.0002 | 9839 |
| Residuals | 74 | 49.845 | 0.6735 | | | |
| Total | 77 | 54.124 | | | | |
| (b) Woody species | | | | | | |
| *Grazing first* | | | | | | |
| Grazing | 2 | 3.1109 | 1.5555 | 5.399 | 0.0001 | 9897 |
| MEM1 | 1 | 1.0529 | 1.0529 | 3.655 | 0.0004 | 9926 |
| MEM2 | 1 | 1.626 | 1.626 | 5.644 | 0.0001 | 9936 |
| MEM3 | 1 | 0.67419 | 0.67419 | 2.340 | 0.0127 | 9926 |
| MEM4 | 1 | 1.5297 | 1.5297 | 5.309 | 0.0001 | 9924 |
| Residuals | 71 | 21.456 | 0.28811 | | | |
| Total | 77 | 28.449 | | | | |
| *Grazing last* | | | | | | |
| MEM1 | 1 | 1.1044 | 1.1044 | 3.833 | 0.0004 | 9918 |
| MEM2 | 1 | 1.915 | 1.915 | 6.647 | 0.0001 | 9928 |
| MEM3 | 1 | 1.6641 | 1.6641 | 5.776 | 0.0001 | 9921 |
| MEM4 | 1 | 2.4528 | 2.4528 | 8.514 | 0.0001 | 9927 |
| Grazing | 2 | 0.85743 | 0.42871 | 1.488 | 0.0791 | 9902 |
| Residuals | 71 | 20.456 | 0.28811 | | | |
| Total | 77 | 28.449 | | | | |

To determine how the abiotic and spatial covariates influenced these differences, we performed a one-way permutational multivariate analysis of variance with grazing levels as a factor, and controlling for covariates derived from the 'best' subset db-RDA models (Tables 2 and 3). For both herbaceous and woody communities, grazing intensity had a significant effect ($p < 0.001$) for herbaceous and woody species for 69 sites, and for herbaceous species for 78 sites, regardless of the order of entry in the model. However, grazing had a marginally significant effect on beta-diversity in woody communities for 78 sites when it was entered last, confirming that was spatially structured.

# DISCUSSION

Our results confirmed that the abiotic environment, grazing, and spatial relationships between sites collectively shaped plant communities at Jebel Ichkeul, although their

**Table 3  Permutational multivariate ANOVA (Hellinger distance) testing herbaceous and woody species composition among grazing classes (crossed design, grazing as fixed effect) at 69 sites with *O. europaea* densities controlling for abiotic and spatial covariates.** (a) Herbaceous; (b) Woody.

| Source | Df | SS | MS | Pseudo F | P (perm) | # unique permutations |
|---|---|---|---|---|---|---|
| (a) Herbaceous species | | | | | | |
| *Grazing first* | | | | | | |
| Grazing | 2 | 2.1619 | 1.081 | 1.575 | 0.0055 | 9833 |
| MEM1 | 1 | 1.6556 | 1.6556 | 2.413 | 0.0004 | 9868 |
| Residuals | 65 | 44.603 | 0.6862 | | | |
| Total | 68 | 48.421 | | | | |
| *Grazing last* | | | | | | |
| MEM1 | 1 | 1.6396 | 1.6396 | 2.389 | 0.0003 | 9871 |
| Grazing | 2 | 2.178 | 1.089 | 1.587 | 0.0066 | 9827 |
| Residuals | 65 | 44.603 | 0.6862 | | | |
| Total | 68 | 48.421 | | | | |
| (b) Woody species | | | | | | |
| *Grazing first* | | | | | | |
| Grazing | 2 | 3.2019 | 1.6009 | 5.296 | 0.0001 | 9904 |
| MEM1 | 1 | 1.0699 | 1.0699 | 3.539 | 0.0004 | 9920 |
| MEM2 | 1 | 1.0382 | 1.0382 | 3.435 | 0.0007 | 9926 |
| MEM3 | 1 | 1.2522 | 1.2522 | 4.143 | 0.0002 | 9920 |
| MEM12 | 1 | 0.9975 | 0.9975 | 3.300 | 0.0007 | 9923 |
| Residuals | 62 | 18.741 | 0.30227 | | | |
| Total | 68 | 26.3 | | | | |
| *Grazing last* | | | | | | |
| MEM1 | 1 | 1.1956 | 1.1956 | 3.955 | 0.0001 | 9924 |
| MEM2 | 1 | 1.3177 | 1.3177 | 4.359 | 0.0001 | 9918 |
| MEM3 | 1 | 2.2144 | 2.2144 | 7.326 | 0.0001 | 9914 |
| MEM12 | 1 | 1.4074 | 1.4074 | 4.656 | 0.0001 | 9934 |
| Grazing | 2 | 1.4246 | 0.71229 | 2.357 | 0.0010 | 9898 |
| Residuals | 62 | 18.741 | 0.30227 | | | |
| Total | 68 | 26.3 | | | | |

influence was strongest in woody communities. However, the individual effect of grazing could not be disentangled from the strong effect of space on overall plant beta-diversity in both the herbaceous and woody communities, potentially reflecting the spatially autocorrelated nature of the grazing regime. Grazing pressure and human activity incorporated using *O. europaea* sizes and densities as a proxy, explained additional beta-diversity only in the woody community types, and had no individual effects on the herbaceous community. Human activity thus appears to exert a slight selective pressure in the woody community overall, though not in its herbaceous counterpart.

Although the overall effect of grazing on beta-diversity was highly confounded with space, grazing intensities drove significant differences in beta-diversity in both the herbaceous and woody communities. Moderate-to-heavy grazing appears to homogenize herbaceous communities, and to a lesser extent, woody communities. Herbaceous communities in
grazed sites, regardless of intensity, differed from ungrazed sites. On the other hand, woody beta-diversity differed only between unbrowsed and moderately-to-heavily browsed sites, suggesting that woody communities were able to withstand light levels of browsing. Grazing's main effect thus appeared to lie in shaping communities according to species' sensitivity to grazing, especially at high intensity levels, although its overall effect on community composition was small.

It is perhaps unsurprising that grazing had little influence on plant community composition in Jebel Ichkeul, given that grazing's effect on community composition and species diversity is scale-dependent and can be challenging to isolate (*Brinkmann et al., 2009*; *Arévalo et al., 2011a*; *Arévalo et al., 2011b*). As space almost universally explained most of the variation in plant beta-diversity, some unmeasured spatially-structured variables might be confounding the effect of grazing. It is also important to note that this study was initially designed to sample floristically diverse communities and thus intentionally neglected the denuded southern border of the mountain, rather than ensuring an even sampling of different grazing regimes. Had we employed a grid sampling design stratified by grazing, the overall effects of grazing on plant community composition might have been clearer.

Despite these sampling issues, we found that different levels of grazing intensity structure compositionally dissimilar communities. Our results also suggested that herbaceous and woody communities responded differently to grazing intensity. Herbaceous communities appeared to be sensitive to both light and moderate to heavy grazing, and became homogenized under higher grazing pressure. On the other hand, woody communities only differed from their unbrowsed state in response to moderate to heavy grazing, suggesting that woody communities were robust to intermediate grazing pressure (*Gabay, Perevolotsky & Shachak, 2006*; *Miguel-Ayanz, García-Calvo & Fernández-Olalla, 2010*). This is consistent with a similar study in the Canary Islands, where woody shrubs declined strongly from abandoned to heavily grazed areas (*Fernández-Lugo et al., 2013*). Assuming that grazing intensity on woody species is representative of grazing pressure on herbaceous species, our findings suggest that management strategies should vary for herbaceous and woody species under grazing regimes. However, we caution that our interpretations could be confounded by differences in species diversity (144 herbaceous species, 35 woody species), and by spatial autocorrelation in grazing pressure and beta-diversity within the two community types.

Although the signal of grazing intensity's effect on plant community composition was weak, this is likely due to limitations imposed by sampling design, categorical grazing pressure measurements, and the assumption of equal grazing on herbaceous and woody communities irrespective of livestock preferences. In fact, plant species on Jebel Ichkeul differ in their responses to grazing, where some species are sensitive to increased grazing pressure, some are maintained by moderate grazing pressure, and some are able to persist under even heavy grazing pressure. This range of responses resulted in a heterogeneous cover of community types along the mountain's grazing gradient. For example, some woody browse species (e.g., *Ceratonia siliqua*, *Coronilla valentina*, *Erica arborea*, *Erica multiflora* and *Rhamnus lycioides*—(*Le Houerou, 1981*) are sensitive to grazing, and were

negatively correlated with the grazing gradient. Instead, the increased humidity on the north side of the lake and deep valley soils allowed the development of closed canopy forest dominated by *O. europaea*, *Ceratonia siliqua* and *Phillyrea angustifolia*. Although these communities are species poor, they are of high ecological value because they are representative of the Mediterranean basin's 'intact' vegetation, presenting an important education and scientific opportunity for researchers and visitors to the national park. Elsewhere in the Mediterranean, the *Olea-Ceratonion* formation is restricted to zones below 300 m (*Tomaselli, 1977*).

Low matorral vegetation, comprised of species with the ability to persist under heavy grazing pressure, was scattered on denuded xeric slopes along the mountain's heavily grazed southeastern perimeter near gourbi villages on the southern face and the western end of the mountain. These highly grazed areas harboured herbaceous indicators of overgrazing, including weedy species (Asteraceae) like *Atractylis cancellata*, *Carthamus lanatus* and *Scolymus hispanicus*, and were lacking many sensitive woody browse species (e.g., *Ceratonia siliqua*, *Coronilla valentina*, *Erica arborea*, and *Rhamnus lycioides*). Even *Pistacia lentiscus*, which is normally resistant to grazing (*Le Houerou, 1981*), was sparse in these areas. These sensitive woody species only occur on the southern slopes of the mountain when sheltered from grazing at higher elevations below the ridge crest, outside the range of goat herds. This overgrazing creates inhospitable conditions for establishment of other protective vegetation on the southern slopes of the mountain, increasing insolation and aridity, and potentially exacerbating this problem.

While overgrazing erodes plant diversity, moderate grazing can maintain openings between woody vegetation patches in some areas of Jebel Ichkeul, removing competing plant biomass, providing nutrients from livestock manure, and ultimately increasing plant species richness. For example, cattle (45–60) and numerous wild boar that foraged in the Jebel Ichkeul created the open conditions which suit Orchidaceae and Liliaceae (*Fay, 1980*), as well as various socio-economically important pasture grass species (*Noy-Meir & Oron, 2001*). Most of the orchid species in the *Ampelodesma* stand at Saida Lalia Hadan, which contains the largest *O. europaea* in the Park, are also maintained by intermediate levels of grazing (*Fay, 1980*). Nevertheless, Gramineae, which include some valuable pasture species, appeared to be excluded from heavily grazed sites in Jebel Ichkeul. The level of grazing needed to maintain the richness of different plant groups should thus be investigated in further research.

Importantly, more knowledge of the herbivore community might have shed more light on the effects of grazing on plant community structure. An estimated 2,500 grazing animals browsed on the mountain in 1980 (*Fay, 1980*; Anonymous, 1998, unpublished data), primarily comprised of goats in addition to sheep and cattle (*Hollis, 1977*). Livestock type affects grazing impacts on vegetation structure and composition (*Tóth et al., 2018*), with goats generally being the most destructive species (*Tomaselli, 1981*). Although the composition and abundance of the herbivore community were not assessed in this study, their effect on the outcomes of grazing merit further investigation in future research.

Translating these findings into conservation and management recommendations requires an acknowledgement of three major changes in human occupation and abiotic

conditions at Jebel Ichkeul since sampling occurred in 1983. First, around 1,000 people lived in the gourbi villages along the southern flanks of the Jebel from 1983 to 2004, but this number dropped to 400 by 2008, declining until these villages were entirely evicted. Although grazing pressure has not been completely eradicated, the change in livestock abundance and distribution is likely substantial. Second, the limestone quarries which were contributing to xeric conditions through dust deposition have been closed down (*UNEP-WCMC & IUCN, 2017*). Third, Tunisia has undergone climate change since at least the 1970s (*Paeth et al., 2009*), and faces a predicted 20% decrease in precipitation and 1° and 3 °C increase in temperature by 2050 (*Tramblay, El Adlouni & Servat, 2013*; *Dakhlaoui et al., 2017*). These changing climatic conditions, in addition to reduced dust deposition and altered grazing regimes have probably substantially altered the relationships investigated in this study. While the dams installed at all of the major rivers flowing into Lac Ichkeul (*UNEP-WCMC & IUCN, 2017*) could have created more xeric conditions, especially on the northern slopes of the Jebel, this effect was likely temporary and minimal due to substantial rainfall between 2003 and 2006, which restored former water and salinity levels.

## CONCLUSIONS AND MANAGEMENT RECOMMENDATIONS

Maintaining the heterogeneity and diversity of vegetation types and structure at Jebel Ichkeul is an important management goal, in order to maintain the region's high ecological value (see *Gabay, Perevolotsky & Shachak, 2006*; *Miguel-Ayanz, García-Calvo & Fernández-Olalla, 2010*). A key step towards this goal is the consistent monitoring of the spatial distribution and structure of vegetation, particularly with regards to rare herbaceous species and woody species. Importantly, our study provides a useful baseline of the plant assemblages at Jebel Ichkeul with which to compare future vegetation changes. Re-surveying plant assemblages at our survey sites could reveal how changes in grazing intensity, climate, and human occupation have affected plant community composition and diversity over several decades. Going forward, vegetation structure could be monitored using a combination of remote sensing and on-the-ground metrics of patch size, patch density, or edge density (*Glasser et al., 2013*). However, ground truthing of species composition, gap dimensions and height of vegetation would be essential, given that grazing pressure is focused below the canopy (*Glasser et al., 2013*), though LiDAR imagery may circumvent this problem.

Recovering the maquis vegetation, particularly on the southern base of the mountain, would protect the watershed by reducing soil erosion and consequent sedimentation of adjacent marshland, and should therefore be a high priority goal. Restricting grazing and woodcutting on the Jebel would directly facilitate this recovery. According to *Fay (1980)*, the spatially heterogeneous grazing pressure is the Jebel Ichkeul park's main conservation and management problem. Managed grazing would be preferential to address this issue (*Arévalo et al., 2012*; *Lázaro et al., 2016*), rather than complete eradication of livestock, which can have unintended consequences such as biological invasions or succession into less desirable plant communities (*Mata et al., 2014*; *Arévalo et al., 2012*). As such,

implementing strategies to control grazing on the southern slopes of the mountain, combined with wood-cutting restrictions, could enable vegetation to re-establish on the degraded southern slopes.

Effective management of the park should balance biodiversity goals with multiple uses, including traditional pastoralism (*Verdú, Crespo & Galante, 2000*; *Perevolotsky, 2005*), to maintain valuable ecosystem services provided by the Jebel within Ichkeul national park (*Daly-Hassen, 2017*). A systematic conservation planning tool could prioritize areas based on maximizing spatiotemporal variation in plant assemblages (*Levin et al., 2013*), effectively maintaining vegetation diversity across the landscape, delineating zones for controlled grazing, and setting biodiversity targets for specific vegetation types.

Given the predicted decreases in precipitation and increases in aridity (*Tramblay, El Adlouni & Servat, 2013*; *Dakhlaoui et al., 2017*) in North Africa, it is vital to consider the role of the Jebel's complex topography and its proximity to Lake Ichkeul in creating microclimate refugia (*McLaughlin et al., 2017*), which can locally buffer climate extremes for many endemic plant species (*Harrison & Noss, 2017*). The park's potential function as a climate refugium solidifies the need to effectively manage grazing and other anthropogenic activities, in order to maintain plant diversity and ecosystem functioning in this vulnerable region.

## ACKNOWLEDGEMENTS

This research was conducted as part of an M.Sc. Thesis in Conservation at University College London by the first author. Special thanks to JM Fay for pioneering this project (a plant inventory of Jebel Ichkeul completed in 1980), and his devotion to protected area conservation in Africa. AC Stevenson of UCL and R Vickery of the British Museum (Natural History) helped with plant species identifications. We would also like to thank JM Fay, B Green, the late GE Hollis, JD Skinner, A Warren and JB Wood for logistical support or advice. BT Collins helped with the equations for olive diameter measurements. We are especially indebted to MJ Anderson for statistical advice, L Olson for help with preliminary variance partititioning, C Fauvelle and C Widdifield for help with GIS, and A Daoud-Bouattour and M Ouali for checking plant nomenclature. We thank A Daoud-Bouattour, RG Gavilan, PW Rundel and AC Stevenson for general comments on earlier drafts of this manuscript. Two anonymous referees provided extremely helpful comments during formal review of the manuscript.

### Funding

This work was supported by the Natural Environment Research Council (UK). Funding for the manuscript for DAK was provided by Ivan and Thelma Kirk. KH was supported by an FQRNT doctoral research scholarship. The funders had no role in study design, data collection and analysis, decision to publish, or preparation of the manuscript.

## Grant Disclosures

The following grant information was disclosed by the authors:
Natural Environment Research Council (UK).
FQRNT doctoral research scholarship.

## Competing Interests

The authors declare there are no competing interests. However, David Anthony Kirk is Chief Executive Officer of, and owns, the ecological research consultancy, Aquila Conservation & Environment Consulting. There is no financial relationship of any kind between the submitted research and the company 'Aquila Conservation & Environment Consulting' or any other organization. The original fieldwork and preparation of the original thesis were funded by the Natural Environment Research Council (UK) and done at University College London in 1983.

## Author Contributions

- David Anthony Kirk conceived and designed the experiments, performed the experiments, analyzed the data, contributed reagents/materials/analysis tools, prepared figures and/or tables, authored or reviewed drafts of the paper, approved the final draft.
- Katherine Hébert analyzed the data, contributed reagents/materials/analysis tools, prepared figures and/or tables, authored or reviewed drafts of the paper, approved the final draft.
- Frank Barrie Goldsmith conceived and designed the experiments, approved the final draft, helped with field design, supervised project, commented on manuscript.

## Data Availability

Kirk, D.A.; Hébert, K.; Goldsmith, F. B. (2019). Plant biodiversity data and environmental and spatial data from Jebel Ichkeul, a limestone mountain in northern Tunisia (1983). NERC Environmental Information Data Centre. https://doi.org/10.5285/4e36cdfa-0281-423f-99a8-e7c331b2e0d1.

## Supplemental Information

Supplemental information for this article can be found online at http://dx.doi.org/10.7717/peerj.7296#supplemental-information.

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
