# Peer review of "Grazing effects on woody and herbaceous plant biodiversity on a limestone mountain in northern Tunisia"

_PeerJ, doi:10.7717/peerj.7296_

## Round 0.1 · original submission · Major Revisions

Both reviewers have provided detailed feedback on how to revise this manuscript and how to better match the outcomes to the data. There is a consensus that the data are worthy of publication but the analysis, results and discussion need further work.

Reviewer 1 ·

Basic reporting

The manuscript provide information on maquis vegetation subjected to various intensity of grazing pressure in Tunisia. The authors suggested that grazing can substantially influence the subjected communities and higher grazing pressure cause a homogenisation (expressed in a decrease of beta diversity) but moderate grazing helps to sustain biodiversity. Based on my evaluation the paper has some shortages which should be treated in a revision. Major point of critique subject sampling setup, quantification of grazing and the consciseness of the ms.

Experimental design

(i) Sampling setup – it is not clear how plots/sites were selected for study, which type of livestock grazing subjected the sites (i.e. it is written briefly that goat, sheep and cattle was grazed there – but this type of grazers cannot be treated equally – goats grazing almost everything, sheep are rather selective for forbs, while cattle grazing is directed to community dominants – i.e. in grasslands mostly to the grasses and are selective for patches with high biomass). See respective very recent case study comparing cattle and sheep grazing Tóth et al. (2018).
Tóth et al. (2018): Livestock type is more crucial than grazing intensity: Traditional cattle and sheep grazing in short-grass steppes. Land Degradation & Development, DOI: 10.1002/ldr.2514
(ii) Quantification of grazing – You mentioned here that the magnitude of grazing pressure was quantified mostly by a the semi-quantitative (ordinal) scale – how high was livestock browsing on trees and shrubs and how high was the presence of other tracks (e.g. fur, droppings). But then you made conclusions on graminoids/forbs of the understory. How could you do that if you haven’t assessed the browsing effects on it at the same scale? It could be that grazing is not subject the trees/shrubs but highly affect the understory. Also it is rather unclear, how you made the assessment that 71% of sites were subjected almost no grazing (group 1). You assessed all schrub and tree species ranking them into the 4 grazing category and finally you made a semi-quantitative mean and if the overall score was below 1.5 than you considered it as not grazed? Some details would be necessary here.

Validity of the findings

No comment. Can be evaluated after the methods section is clear.

Additional comments

(iii) The ms is far too long. I suggested below shortening for the intro, but other sections of the ms also should be shortened by at least 20%. Too wordy and overloaded – but not for the really necessary information.

l59-65 You should also consider rich research in grassland grazing – which has quite similar outcome to your expectations for maquis vegetation. See for example Török et al. (2016) special issue in AGEE for grazing and papers therein.

Török et al. (2016): Grazing in European open landscapes: how to reconcile sustainable land management and biodiversity conservation? Special Issue in Agriculture, Ecosystems and Environment.
https://www.sciencedirect.com/journal/agriculture-ecosystems-and-environment/vol/234

l103-110 Can be omitted.
l111-129 Shorten and move some sections concerning site description to the Materials and methods section
l133 What beta diversity do you refer here (I assume Bray-Curtis similarity)
l136-140 Can be omitted.
l181-192 should be more precise – see general comments – grazing animals, site selection – and maybe a figure on the sampling setup would be fine.
l221-222 Not considered forbs and graminoids in grazing intensity measures. How were grazing effects (i.e. different levels of grazing on subjected species – there is generally preferred and non-preferred species with different levels/magnitude of grazing)
l235 Provide details on altimeter or skip.
l270 which livestock type, was it subjected different type at different site(s).
l400-404 If you do not provide information on vegetation differences, site heterogeneity we cannot assume that the sites were different because of different levels of grazing…

Reviewer 2 ·

Basic reporting

The study examines the effect of grazing and other environmental factor on the vegetation composition of an area in southern Mediterranean region. Given that this region is rather underrepresented in the relevant literature regarding the ecology of Mediterranean ecosystems, the data presented in the manuscript are interesting and worth publishing in an international Journal such us PeerJ. The manuscript reads well and academic English is used throughout. Some spelling and typing errors does not change the proper use of English language. The scopus of the study falls within the interests of the Journal. However, there are several issues that need to be addressed before the manuscript is considered for publication.
The introduction is well written, well structure and well referenced providing all the necessary background information to the reader which allows him to understand the historical role of humans in shaping the Mediterranean landscape, and the need to reconcile anthropogenic activities with nature conservation under the frame of climate and global change. Perhaps it is too extensive and it could be shortened without missing vital information, but I do not consider it an issue for an open access journal.

Experimental design

The sampling design was originally designed for a floristic inventory rather than the study of grazing intensity effect. This resulted in an uneven distribution of the grazing classes within the samples. Furthermore, given the higher sampling intensity of the northern slopes I assume that the most intensively grazed areas were avoided as they probably were too degraded. The uneven representation of the grazing classes within the sample, results in several issues regarding the conclusion that can be drawn based on the multivariate analysis performed in the manuscript. Furthermore, given the high sampling intensity I assume there might be several spatial autocorrelation issues among the samples which the author did not account for. This must be especially true for the low intensity grazing class. Perhaps the authors should run an autocorrelation analysis first and omit some samples which show high spatial autocorrelation. This would result to a smaller number of samples for the further analysis but perhaps more evenly distributed among the grazing intensity classes.
The authors use an indirect method of measuring grazing intensity which relies on the morphological characteristics of shrubs. Although this method does not provide accurate estimation of grazing intensity, compared to other methods such as pellet or animal counts, it is appropriate for such study since it is able to capture the log term effect of grazing on vegetation structure. This is particularly true for woody species which does not respond as quickly as herbaceous species to changes in the grazing regime.
In lines 268-279 the authors describe the way data related to human footprint were collected. It is not clear to me how road data were collected based on OpenStreetMap dating back to 1982. Furthermore, proximity to roads and settlements is often used as a proxy of grazing intensity since in eastern Mediterranean where extensive husbandry is still practiced the herds are mostly kept around settlements. Thus, those two experimental variables when they are accounted for it is expected that the effect of grazing will be minimised. Distance from roads and settlements have only an indirect effect on vegetation through the severity of human activities including wood cutting, grazing and fire. So including them as covariables is likely to obscure the effect of grazing and other anthropogenic activities.
The way environmental variables are treated in order to avoid anomalies in their distribution and collinearity is sound and appropriate and so are the statistical methods employed.

Validity of the findings

The results presented in lines 390-399 are rather confusing. According to the authors the nMDS analysis reveal a clear effect of grazing but at the same time they imply that the grazing has a clear altitudinal pattern. If this is the case how the effect of grazing is separated from this of altitude. In my view perhaps axis 1 represents a grazing intensity gradient, but it is difficult to sai that with certainty because no clear clustering occurs and the high intensity grazing sites fall along the first axis between the no grazing and low intensity grazing sites. Axis 2 seem not to be related to grazing. With that being said I don’t think that the claim made by authors that Ceratonia siliqua is an indicator of unbrowsed sites is justified by this ordination plot. Figure 3 again is not able to give some indication or hypothesis on the potential role of grazing in species composition since again no clear clustering of samples occur. I believe to a great extent that this occurs due to the uneven sample sizes between the 4 grazing regime classes.
In lines 408 to 427 the authors present the results of a modelling approach based on dbRDA. The two ordination plots (4a and b; reference to 4b is missing in the text) and the presented results seem to underestimate the effect of grazing for woody plants (given that it rarely appears in the models). However, I believe that the way geographic location is treated here which according to authors represent a proxy of humidity might confound the effect of other factors including grazing. I am not sure geographic location is an appropriate proxy for humidity and a valid explanatory variable and the results I think they justify my previous comment that an autocorrelation analysis is needed beforehand.
In line 439 why the authors do not accompany their result with a figure as part of figure 5? In lines 450-456 the authors state that there is a significant effect of grazing in the herbaceous species richness and diversity but Supplementary Figure 3 suggests otherwise. There is a discrepancy there that needs to be clarified. Furthermore I believe Supplementary figure 3 should be included in the main manuscript for the woody plants where significant effects are found.
In lines 460-468 the authors mention in the text tables 4a and b but only one table 4 seem to exist in the tables. Figures 7c and d are not mentioned in the text and the caption is incomlpete.
Overall it seems to me that grazing had a minor effect in determining the vegetation structure and composition in the study area which is rather surprising especially for the beginning of 80s. I also believe, as the authors do, this has to do with the way the grazing intensity classes are represented in the sample and a careful filtering of the samples based on spatial location and autocorrelation issues would reveal a much more informative picture regarding the effect of grazing and other factors.
Apart from the first part of the discussion, in the rest various issues relate to the ecology of Mediterranean ecosystems are discussed which are not directly related to the presented results. However, it is very well written and interesting and I believe the authors could change the scopus of the study into a more generic investigation of factors affecting vegetation structure and composition in the area, and focus on some specific species and assemblages of conservation and ecological importance much of those discussed in the manuscript would be in accordance to the presented results.

Additional comments

Overall I would like to encourage the authors to elaborate more on their data, alter the aim and objectives of the study and revise the manuscript because I sincerely believe that there is important knowledge that can be generated by this dataset despite its age.

---

## Round 0.2 · accepted · Accept

Thank you for embracing the feedback from the original reviews. I am entirely happy that you have addressed the issues raised and consider it important to get this paper into the public domain now.

Reviewer 2 ·

Basic reporting

No comment

Experimental design

No comment

Validity of the findings

The manuscript contribute significantly to the scientific literature regarding the ecology and dynamics of Mediterranean ecosystems. This contribution becomes even more significant considering that it regards a geographic region rather underrepresented in the literature and vulnerable to the foreseen climate and other socioeconomic changes.

Additional comments

Dear Editor and Authors.
I am glad to have contributed to the substantial improvement of the manuscript. All my comments have been properly addressed by the authors in the revised version of the manuscript. With that being said I have no hesitation to recommend the publication of the manuscript and I am looking forward to see it published in PeerJ.